

# Expression and phylogeny of multidrug resistance protein 2 and 4 in African white backed vulture *(Gyps africanus)*

Bono Nethathe[1,2], Aron Abera[3] and Vinny Naidoo[1]

[1] Department of Paraclinical Science, Faculty of Veterinary Science, University of Pretoria, Onderstepoort, Pretoria, South Africa
[2] Department of Food Science and Technology, University of Venda, Thohoyandou, Limpopo, South Africa
[3] Inqaba Biotechnology, Sunnyside, Pretoria, South Africa

## ABSTRACT

Diclofenac toxicity in old world vultures is well described in the literature by both the severity of the toxicity induced and the speed of death. While the mechanism of toxicity remains unknown at present, the necropsy signs of gout suggests primary renal involvement at the level of the uric acid excretory pathways. From information in the chicken and man, uric acid excretion is known to be a complex process that involves a combination of glomerular filtration and active tubular excretion. For the proximal convoluted tubules excretion occurs as a two-step process with the basolateral cell membrane using the organic anion transporters and the apical membrane using the multidrug resistant protein to transport uric acid from the blood into the tubular fluid. With uric acid excretion seemingly inhibited by diclofenac, it becomes important to characterize these transporter mechanism at the species level. With no information being available on the molecular characterization/expression of MRPs of *Gyps africanus*, for this study we used next generation sequencing, and Sanger sequencing on the renal tissue of African white backed vulture (AWB), as the first step to establish if the MRPs gene are expressed in AWB. In silico analysis was conducted using different software to ascertain the function of the latter genes. The sequencing results revealed that the MRP2 and MRP4 are expressed in AWB vultures. Phylogeny of avian MRPs genes confirms that vultures and eagles are closely related, which could be attributed to having the same ancestral genes and foraging behavior. In silico analysis confirmed the transcribed proteins would transports anionic compounds and glucose.

# INTRODUCTION

In India, birds belonging to the *Gyps* genus (Oriental white-backed vulture, *Gyps bengalensis*; long-billed vulture, *G. indicus* and slender-billed vulture, *G.tenuirostris*) were in grave danger of extinction in the 1990s following their inadvertent exposure to veterinary diclofenac that found its way into their food chain (*Oaks et al., 2004*). Following their single exposure to quantities of drug that equated to a dose in the region of 0.8 mg/kg, birds were found dead within 48 h of said exposure with signs of severe visceral gout and associated

Corresponding author
Bono Nethathe,
bono.nethathe@univen.ac.za

renal and hepatic damage (*Oaks et al., 2004*; *Green et al., 2004*; *Shultz et al., 2004*; *Swan et al., 2006*). Despite the diclofenac induced deaths being largely recognized as one of the worst environment intoxications in recent times, the mechanism behind the evident gout is still incompletely understood. One of the theories put forward, suggests that toxicity is due to the inhibition of the uric acid transporter channels in the renal tubular epithelial cells (*Naidoo et al., 2007*), which would be consistent with diclofenac inhibition of uric acid transport in mammals (*Khamdang et al., 2002*; *Nozaki et al., 2007*; *Burckhardt, 2012*).

Multidrug resistance proteins (MRPs) are part of the uric acid transporters that play an important role in the excretion of uric acid from the intracellular environment, following their movement therein by the organic anionic transporters in mammals (*Sweet, 2005*; *Sweet, 2010*; *VanWert, Gionfriddo & Sweet, 2010*). The MRP family also known as ATP-binding cassette subfamily C (ABCC) was first described as a drug resistance protein by *Cole et al. (1992)* when they managed to clone the MRP1 gene in human (*Cole et al., 1992*). They first associated it with transporting anti-cancer drugs (*Haimeur et al., 2004*). MRP1 is generally regarded as the godfather of the family and further research has since described five homologs named MRP2-6 (*Kool et al., 1997*; *Kool et al., 1998*; *Kool et al., 1999*).

In general, the MRP protein functions as an extrusion pump system responsible for moving substances from within the cell into the extracellular environment e.g., in biliary transport it moves substances into the biliary tract and bile (*Taniguchi et al., 1996*; *Kool et al., 1997*), while MRP2 expressed in the proximal renal tubule endothelial cells found in apical membrane plays a role in the excretion of small intracellular organic anions (*Sekine, Miyazaki & Endou, 2006*). Examples of substance transported by renal MRP2 are lipophilic substances conjugated to glutathione, glucuronate, or sulfate (*König et al., 1999*). The MRP transport proteins are distributed throughout the body with MRP3 and MRP4 found in the pancreas, bladder, gut, lung, prostate, ovary, muscle, testis, kidney and gallbladder respectively while MRP5 being ubiquitous (*Kool et al., 1997*; *Kool et al., 1999*; *Lee et al., 1998*; *Kiuchi et al., 1998*; *König et al., 1999*). Nonetheless, the main MRP transporters are located in the liver and kidney.

With diclofenac having the ability to induce severe hyperuricemia and gout in the vultures (*Naidoo & Swan, 2009*), this study focuses on characterizing the last step in uric acid transport. More specifically, we focused on the molecular characterizing of the MRP2 and MRP4 transporter in the kidney of G*yps africanus* (African white backed) vulture. In addition to being known to transport organic anions including uric acid, it is believed that these channels function in the same manner as in mammals. The rationale for this approach, it that once the channel has been characterized, more specific expression and cloning studies can be undertaken on these channels. While no such studies have been undertaken in an avian species, studies on cloned MRP transporter proteins from humans have been successfully used to evaluate the effect of specific non-steroidal anti-inflammatory drugs as inhibitors of these transporters (*El-Sheikh et al., 2007*). Moreover we compare the results to the recent study by *Barik et al. (2019)* which characterized MRP4 gene on *Gyps Himalyanesis* for comparison to results with obtained in this study (*Barik et al., 2019*).

## MATERIAL AND METHODS

### Sample collection

Before the commencement of experiments, ethical clearance for collection of samples was conducted according to the guidelines approved by University of Pretoria Animal Ethics committee (V108-16). Two African white-backed (AWB) (*Gyps africanus*) vultures of unknown age with a badly fractured wing and had not been treated with any analgesic previously were euthanized at Faculty of Veterinary Science, University of Pretoria. The cranial lobe part of the kidneys was harvested immediately after euthanasia and stored in cryogenic vials containing RNAlater (Whitehead Scientific, Cape Town, South Africa) and stored at −80 °C freezer until analysed.

### Next generation sequencing (NGS)

Total RNA was extracted from AWB1's kidney using the RNeasy plus mini kit (Qiagen) according to the manufacturer's instructions and transported to Agricultural Research Council (ARC, Onderstepoort, Pretoria, South Africa) on dry ice for sequencing. DNA contamination was removed from total RNA using DNase. The RNA fragments were reverse transcribed into cDNA and sequencing adaptors were ligated. Finally, the ends of the cDNAs were sequenced using Illuminia Truseq mRNA stranded Ran preparation kit on Hiseq 2500 v4 2 × 125 bp chemistry model. The result obtained from sequencing was 50 million short reads which were 125 nucleotides long each. Prior assembly of the raw reads, several quality control steps were conducted on the raw reads using the online Galaxy platform (*Blankenberg et al., 2010*) and fast quality check (FASTQC) (*Andrews, 2014*) to check overall quality of the sequences, duplication and adapters. After checking the quality of the sequences, the adapters and PCR duplicates were removed using the programme Trimmomatic (*Bolger, Lohse & Usadel, 2014*). The pre-processed reads were assembled into transcripts using TRINITY (*Grabherr et al., 2011*). The assembled transcriptome was converted to a local blast database NCBI (*Altschul et al., 1990*) and sequences were identified based on the predicted MRP2 (XM_030031380.1) and MRP4 (XM_030036470.1) sequences from the Golden eagle genome. The Golden Eagle was selected as another study completed in our laboratory has shown them to be closely related species as well as having both channels present. Transcript quantification was undertaken using Sailfish with reference transcripts from the de novo analysis. Alignment of AWB MRP4 and *Gyps Himalyanesis* was also conducted using Clusta omega to reveal similarity.

### Confirmation of transcriptome MRP2 and MRP4 using Sanger sequencing

Fresh total RNA was extracted from AWB2's kidney using Quick RNA Miniprep kit following instructions from the manufacturer (Zymo Research, USA). cDNA synthesis was performed using Lunascript RT supermix kit from New England Biolabs (NEB), (USA) according to the manufacturer's instructions. Primers of MRP2 and MRP4 were designed based on the transcriptome sequences generated from illuminia next generation sequencing. The special primers were synthesized at Inqaba Biotechnology, Pretoria, South Africa. Using the latter set of primers, PCR was performed using One taq polymerase kit

from NEB and AWB's kidney cDNA as a template. The amplification protocol for all the targets was as follows: Initial denature 94 °C 30 s, 40 cycles of (denature 94 °C for 30 s; annealing 55 °C for 30 s and elongation 68 °C for 2 min) and final elongation 68 °C 5 min. However for MRP2 segment 1, the former did not work so the following cycling conditions were used, Initial denature 98 °C 30 s, 40 cycles of (denature 98 °C for 30 s; annealing 65° C for 30 s and elongation 72 °C for 2 min) and final elongation 72 °C 2 min. The PCR products was run on 1% agarose gel. The gel was viewed for desired bands using gel documentation system and PCR products of the desired bands was further cleaned up using Exopsup from NEB and sequences were generated using ABI 3500X1 genetic analyser.

## Phylogenetic analyses

After confirming the NGS data with Sanger sequencing, searches for homologous sequences were performed using BLASTn (*Altschul et al., 1990*). Thirty-five (MRP2) and 36 (MRP4) genes that were 85% and above identical to NGS sequences were downloaded from GenBank (NCBI). A multiple sequence alignment was performed along with the downloaded sequences, using muscle algorithm (Molecular Evolutionary Genetics Analysis version X (MEGA X)) software package (*Edgar, 2004*; *Kumar et al., 2018*). The evolutionary history was inferred by using the Maximum Likelihood method and Tamura-Nei model for MRP2 (*Tamura & Nei, 1993*) while MRP4 was Maximum Likelihood method and General Time Reversible model (*Nei & Kumar, 2000*). The bootstrap consensus tree inferred from 1000 replicates was taken to represent the evolutionary history of the taxa analyzed (*Felsenstein, 1985*). Branches corresponding to partitions reproduced in less than 50% bootstrap replicates were collapsed. The percentage of replicate trees in which the associated taxa clustered together in the bootstrap test (1000 replicates) are shown next to the branches (*Felsenstein, 1985*). Evolutionary analyses were performed using MEGA X (*Kumar et al., 2018*).

## In silico analysis

The obtained MRP2 and MRP4 NGS sequences were converted in Expasy (*Gasteiger et al., 2003*) to protein sequences for their open reading frame. The deduced amino acids sequences were analysed by following software; Transmembrane helix prediction using a Markov model (*Sonnhammer, VonHeijne & Krogh, 1998*), Swiss model Protparam for functional prediction of the protein (*DeCastro et al., 2006*) and PHYRE2 (*Kelley et al., 2015*) for prediction of 2-D and 3-D structures. PROTTER sequence database was used to predict N-glycosylation sites (*Omasits et al., 2013*) and Expasy database was also used to examine other possible post-glycosylation and phosphorylation. The Walker A, ABC signature motif and Walker B segments which are hallmarks of human MRP 2 and 4 were evaluated for its presence (*Hipfner, Deeley & Cole, 1999*; *Choudhuri & Klaassen, 2006*), to conclude that the proteins were functional MRP. This was compared for both nucleotide binding domain 1&2 (*Ren et al., 2004*).

## RESULTS

### Next generation sequencing

The assembled transcriptomes was compared to the golden eagle MRP2 (XM_030031380.1) and MRP4 (XM_030036470.1) sequences. The predicted AWB MRP2 (MN691108) and MRP4 (MN691109) genes were submitted to NCBI and consisted of 5212 and 4061bp respectively. The MRP2 and MRP4 alignment of the AWB vulture and the golden eagle were 98.20% and 99.16% similarity respectively. Oddly the AWB and *Gyps Himalyanesis* (KX168697.1) alignment of MRP4 gene revealed 94.06% similarity.

### Sanger sequencing of data

To confirm the above NGS results, Sanger sequencing was conducted. The cDNA from the AWB kidney was used as template. MRP2 was amplified into two segments using primers design based on NGS data. The PCR products were viewed on a gel under UV documentation and it revealed product size of 2,754 bp and 2,550 bp for segment 1 and 2 respectively (Fig. S1). Moreover amplification of MRP4 was also divided into two segments and PCR products were 2,241 bp and 2,136 bp for segment 1 and 2 in that order. The PCR products were sequenced and the sequences were analysed for consensus which revealed 4,537 bp for MRP2 and 3,897 bp for MRP4. While full coverage was not possible, based on the degree of coverage and similarity between NGS and Sanger of 99.76% and 99.43% for MRP2 and MRP4 respectively was observed. The similarity gave us confidence that the NGS sequences are a true representative of the MRPs sequences. These sequences could not deposited in the NCBI as a result of stop codons within the sequences, which we believe were sequencing errors.

### Phylogenetic analysis

The AWB predicted genes were further analysed to confirm its homogeneity compared to the MRP gene of other avian species. The phylogenetic tree topology revealed 2 distinct clades for the vulture and chicken with the golden eagle, bald eagle and AWB vulture belonging to the same clade. The percentage similarity between vulture, eagles and chicken is stated on the internal nodes numbers representing the percentage of 1,000 replicates for which the same branching patterns were attained (Figs. 1 and 2).

### In silico predictions

The predicted AWB MRP2 sequence generated open reading frame (ORF) containing 1,552 amino acids while MRP4 ORF contained 1,287 amino acids with a predicted molecular weight 173,610.3 and 145,113.90 Daltons respectively. Scan prosite results revealed that the above proteins are likely ABC transporter integral membrane type 1 fused domain (ABC-TMF1) and ABC transporter 2. Secondary structure prediction with PHYRE2 revealed that the predicted MRP2 amino acids contain 62% of alpha-helix, 7% of beta-strand and 30% transmembrane helix. While MRP4 PHYRE2 results showed the former to contain 61% alpha-helix, 9% of beta- helix and 26 to be coiled in nature. TMHMM software analysis for MRP2 revealed the presence of two nucleotide binding domains (NBDs) and two transmembrane domains (TMDs) consisting of 6 and 5 transmembrane

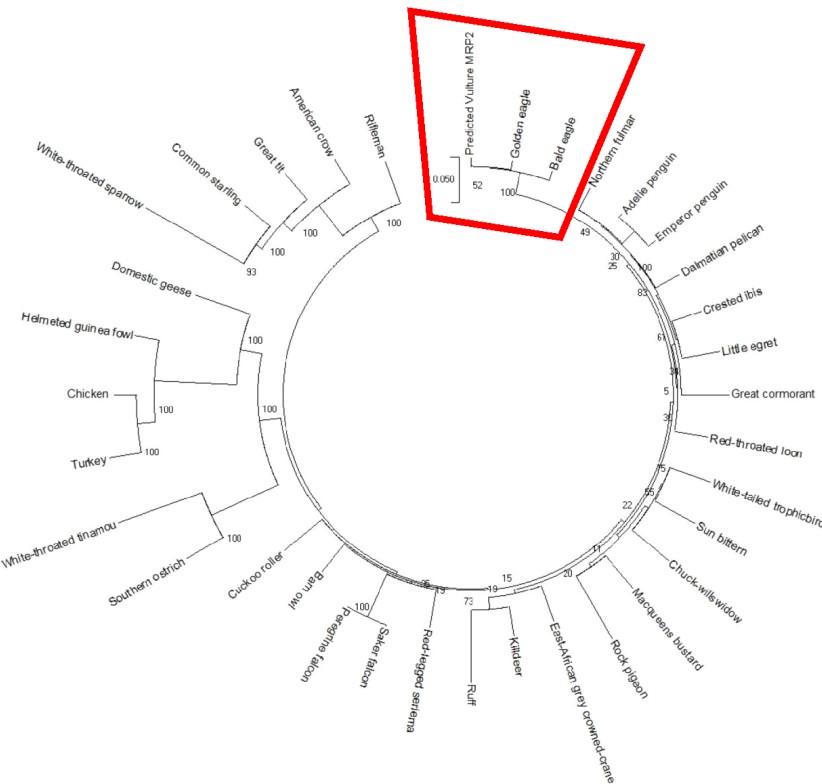

**Figure 1** **MRP2 phylogenetic tree.** Reconstruction of phylogenetic relationship between MRP2 genes of different avian species by maximum likelihood best model: bootstrap.

helices (TMH) respectively with an extra transmembrane domain (TMD) with 5 TMH and for MRP4 it revealed their presence of two NBDs and two TMDs containing 5 and 6 TMH each. PROTTER sequence database was used for prediction of N-glycosylation sites for MRP2 and were found at position 15, 106, 284, 551, 620, 725, 773, 886, 1,019, 1,193, 1,254, 1,383, 1,443 and MRP4 sites were at position 428, 706, 716, 1,009, 1,138 for both AWB vulture and golden eagle moreover confirming the above results (Fig. 3). The quaternary structures predicted by PHYRE2 from MRPs amino acids revealed the presence of two NBDs for both MRPs protein (Fig. 3). The Walker sequence and ABC signature motif are presented in Fig. 4. For both MRP 2 and 4, these sequences were present with one amino acid difference, when a difference was present, in comparison to human MRP2&4.

## DISCUSSION

Previous studies revealed that the *Gyps* species found dead in India was due to the presence of residues of diclofenac in the carcass they fed upon (*Oaks et al., 2004*), which caused signs of abnormally high level of plasma uric acid prior to death (*Naidoo & Swan, 2009*). With uric acid excretion primarily occurring in the kidneys, it has been speculated that diclofenac inhibited uric acid excretion within the renal tubules. The one shortcoming of this hypothesis is that the mechanisms of uric acid excretion is not yet completely
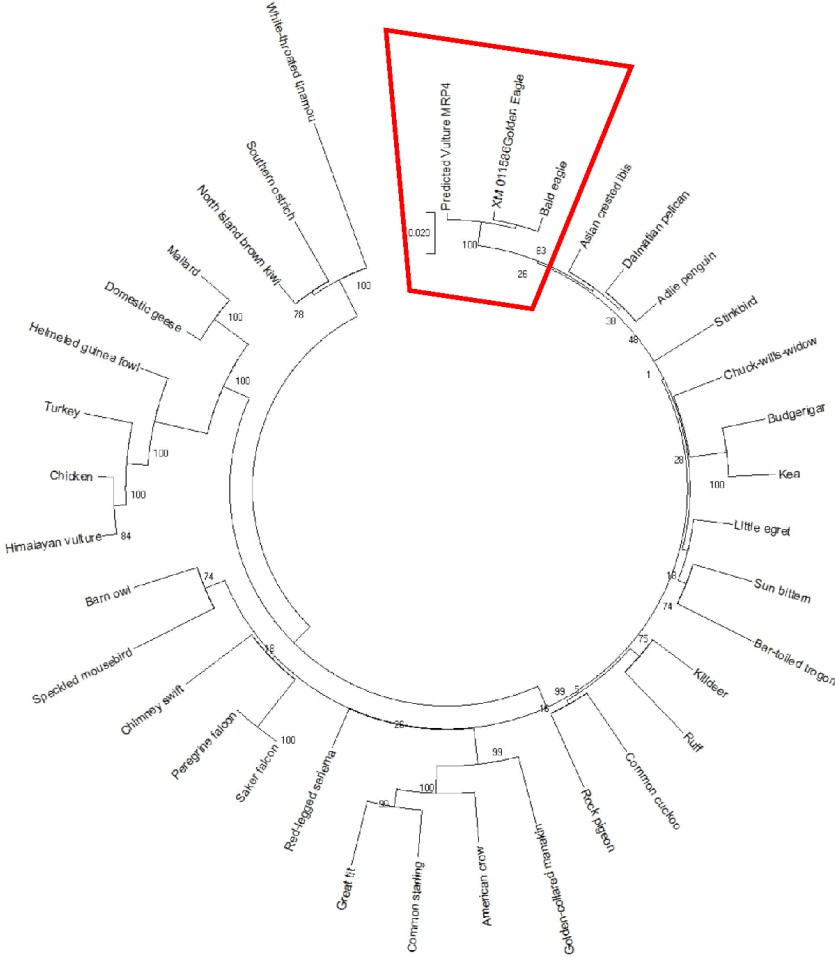

**Figure 2** **MRP4 phylogenetic tree.** Reconstruction of phylogenetic relationship between MRP4 genes of different avian species by maximum likelihood best model: bootstrap.

understood in vultures. Thus to understand the mechanism of toxicity of diclofenac, the mechanism of uric acid excretion needs to be elucidated.

From studies in the chicken, it is known that uric acid in the plasma is excreted at the level of the renal tubules by the organic anion transporters which transports uric acid from the plasma into the cell; while subsequent transport from within cell into the renal tubule being mediated by the MRP transporters (*Sweet, 2005*; *Sweet, 2010*; *VanWert, Gionfriddo & Sweet, 2010*). For this study we focused on the MRP2 and MRP4 transporters, since *Haritova, Schrickx & Fink-Gremmels (2010)* and *Bataille, Goldmeyer & Renfro (2008)*, were respectively able to demonstrate the presence of MRP2 and MRP4 in the chicken kidney. More specifically, we used the AWBV as our test species, which as a vulture species was the second species demonstrated to be highly sensitive to the toxic effects of diclofenac, and more so under controlled laboratory conditions by *Swan et al. (2006)*. In a later rebuttal, *Cuthbert et al. (2007)* indicated that a sample size of two birds was sufficient to demonstrate toxicity due to the reported $LD_{50}$ in other vulture species. Also in a subsequent study, when

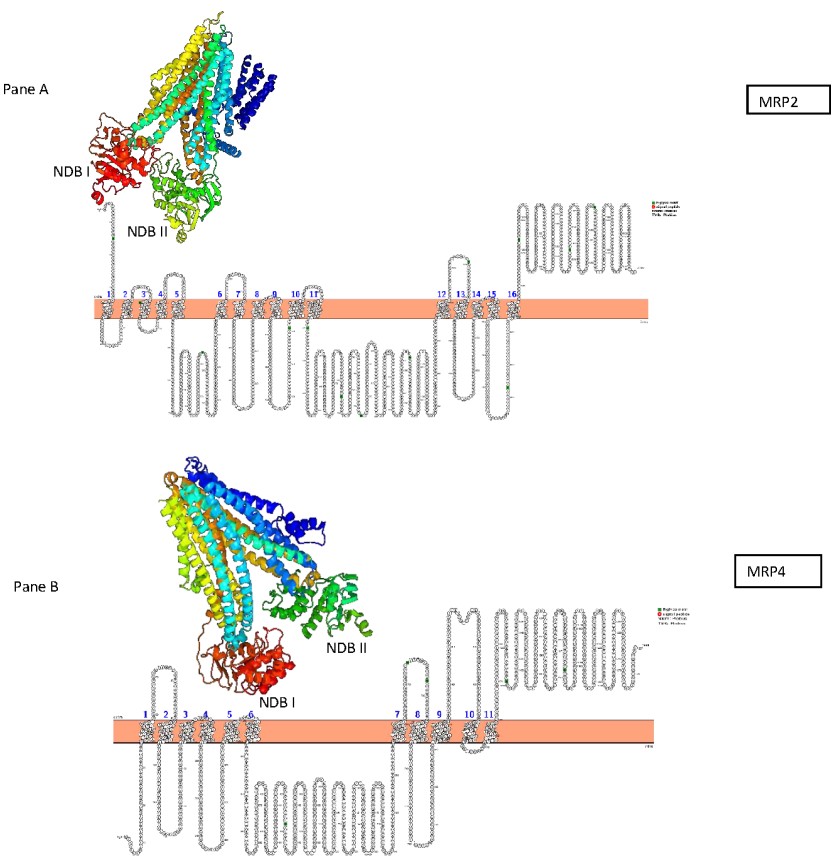

**Figure 3** **Presentation of transmembrane helices with N-gyclan motif sites and 3-D structures.** Pictorial presentation of transmembrane helices and N-gyclan motif sites from (A) MRP2 containing 13 N-glycan motif sites with 16 TMH and (B) MRP4 comprising of 5 N-glycan motif sites with 11 TMH. Also shown are the corresponding 3-D structures of MRP2 and MRP4, illustrated by the PHYRE2 software, showing the presence of two NDBs.

the diclofenac plasma concentration was subjected to non-compartmental analysis by *Naidoo & Swan (2009)*, they were able to demonstrate that toxicity in the study vulture species was associated with a very long half-life of 11 and 22 h in the two birds.

For this study both the study methods (NGS and Sanger sequencing) were able to demonstrate that the MRP2 and MRP4 transporters were expressed in the AWB vulture's kidney. With no primers being described in the study species, that we were aware off, led to our first step of using NGS. NGS as a tool is known to sequence hundreds to thousands of genes at one time without the need for specific primers. The latter method provides the power to discover uncommon or novel variants through deep sequencing (*Alekseyev et al., 2018*). In order to verify the results obtained from NGS, Sanger sequencing was subsequently carried out since it is the golden standard for sequencing (*Sanger & Coulson, 1975*). As for *Swan et al. (2006)* we also relied on the use of two AWBV. While the sample size is low, the high sensitivity of the species to toxicity suggests a highly conserved mechanism of uric acid excretion. Also, due to the endangered nature of the species under

```
Nucleotide Binding Domain 1 (NBD1)
         1                                                            60
hMRP2    AGQLVAVIGPVGSGKSSLISAMLGEMENVHGHITIKGTTAYVPQQSWIQNGTIKDNILFG    722
hMRP3    KGALVAVVGPVGCGKSSLVSALLGEMEKLEGKVHMKGSVAYVPQQAWIQNCTLQENVLFG    712
hMRP4    PGELLAVVGPVGAGKSSLLSAVLGELAPSHGLVSVHGRIAYVSQQPWVFSGTLRSNILFG    494
hMRP5    EGKLVGIQGSVGSGKTSLISAILGQMTLLEGSIAISGTFAYVAQQAWILNATLRDNILFG    646
vMRP2    PGSLVAVVGAVGSGKSSLVSAMLGEMENIKGHINIQGSLAYVPQQAWIQNATLKDNILFG    735
vMRP4    RGELLAVIGPVGAGKSSLLSAVLGELPKDKGLINVTGRIAYVSQQPWVFSGTVRSNILFD    454
                  Walker A

hMRP2    TEFNEKRYQQVLEACALLPDLEMLPGGDLAEIGEKGINLSGGQKQRISLARATYQNLDIY    782
hMRP3    KALNPKRYQQTLEACALLADLEMLPGGDQTEIGEKGINLSGGQRQRVSLARAVYSDADIF    772
hMRP4    KKYEKERYEKVIKACALKKDLQLLEDGDLTVIGDRGTTLSGGQKARVNLARAVYQDADIY    554
hMRP5    KEYDEERYNSVLNSCCLRPDLAILPSSDLTEIGERGANLSGGQRQRISLARALYSDRSIY    706
vMRP2    SELDEARYQQVIKACALLPDLELLPAGDQTEIGEKGINLSGGQKQRVSLARAVYSNADIY    795
vMRP4    KEYEKEKYEKVLKVCALKKDLELLANGDLTVIGDRGATLSGGQKARVNLARAVYQDADIY    514
                                             Signature Motif

hMRP2    LLDDPLSAVDAHVGKHIFNKVLGPNGLLKGKTRLLVTHSMHFLPQVDEIVVLGNGTIVEK    842
hMRP3    LLDDPLSAVDSHVAKHIFDHVIGPEGVLAGKTRVLVTHGISFLPQTDFIIVLADGQVSEM    832
hMRP4    LLDDPLSAVDAEVSRHLFELCIC--QILHEKITILVTHQLQYLKAASQILILKDGKMVQK    614
hMRP5    ILDDPLSALDAHVGNHIFNSAIR--KHLKSKTVLFVTHQLQYLVDCDEVIFMKEGCITER    764
vMRP2    ILDDPLSAVDAHVGKYLFEHVLGPKGLLQKKTRILVTHSISFLPQVDNIVVLVAGTVSEH    855
vMRP4    LLDDPLSAVDAEVGRHLFEKCIC--QALHQKISVLVTHQLQYLRAANQILILKDGKMVGK    572
          Walker B

Nucleotide Binding Domain 2(NBD2)

hMRP2    FNNYQVRYRPELDLVLRGITCDIGSMEKIGVVGRTGAGKSSLTICLFRILEAAGGQIIID    1361
hMRP3    FRNYSVRYRPGLDLVLRDLSLHVHGGEKVGIVGRTGAGKSSMTLCLFRILEAAKGEIRID    1350
hMRP4    FDNVNFMYSPGGPLVLKHLTALIKSQEKVGIVGRTGAGKSSLISALFRLSEP-EGKIWID    1101
hMRP5    FENAEMRYRENLPLVLKKVSFTIKPKEKIGIVGRTGSGKSSLGMALFRLVELSGGCIKID    1254
vMRP2    FVDYKVRYRPELELVLQGITCNIGSTEKVGVVGRTGAGKSSLTICLFRVLEAAGGTIIID    1368
vMRP4    FENVNFTYSLDGPLVLRHLSVLIKPKEKVGIVGRTGAGKSSLIAALFRLAEP-EGRIWID    1063
                                             Walker A

hMRP2    LQLGLSHEVTEAGGNLSIGQRQLLCLGRALLRKSKILVLDEATAAVDLETDNLIQTTIQN    1481
hMRP3    QPAGLDFQCSEGGENLSVGQRQLVCLARALLRKSRIIVLDEATAAIDLETDNLIQATIRT    1470
hMRP4    LPGKMDTELAESGSNFSVGQRQLVCLARAILRKNQILIIDEATANVDPRTDELIQKKIRE    1221
hMRP5    LPLKLESEVMENGDNFSVGERQLLCIARALLRHCKILILDEATAAMDTETDLLIQETIRE    1374
vMRP2    LPEGLLHLVSEAGENLSVGQRQLVCLARALLRKAKILILDEATAAVDLETDHLIQTTIRS    1488
vMRP4    LPNKMEMQLAESGSNFSVGQRQLVCLARAVLKKNRILIIDEATANVDPRTDEFIQKTIRE    1183
                  Signature Motif                 Walker B
```

**Figure 4** **Walker sequence and ABC signature motif.** Comparison of the Walker A, ABC Signature motif and Walker B sequence of the first nucleotide binding domain of the vulture MRP2 and MRP4 in comparison on human MRP (the protein sequences used with their GenBank accession numbers were: MRP2, NP_000383.2; MRP3, AAD02845.1; MRP4, NP_005836.2; MRP5, NP_005679.2).

study, it is extremely difficult to obtain fresh tissue for analysis i.e., fresh tissue is only obtainable when birds are euthanised for ethical reasons. This study also relied on *in silico* analysis to predict the structure and functions of the proteins sequenced as a first step for cloning and functional studies.

In silico analysis with Expasy revealed that the ORFs of MRP2 and MRP4 proteins consisted of 1,552 and 1,287 amino acids respectively for both the AWB vulture and golden eagle. This finding is similar to human MRP2 and MRP4 proteins which comprises of 1,545 and 1,325 amino acids respectively (*Borst et al., 2000*). Moreover a recent study by (*Barik et al., 2019*) also revealed that *Gyps Himalyanesis* MRP4 consisted of 1,349 amino acids while Expasy analysis we undertook for the *Gallus gallus* MRP2 (accession number XM_015288821.1) and MRP4 (accession number NM_001030819) consisted of 1,550 and 1,330 amino acids respectively. While the similarity in the above protein sequences of MRP2 and MRP4 in size and length maybe revealing that their structures are conserved,
the phylogenetic tree only showed a 100%, similarity between vulture and eagle for both MRP2 and MRP4 channels, while the chicken belongs to separate clade. This was in accordance with previous findings by *Zhang et al. (2014)* who explored macroevolution. In their evaluation using the full genomes of 48 avian species representing all main extant clades, their phylogenetic tree revealed that vulture and eagles share the same clade distinct from the chickens. *Jarvis et al. (2014)* and his team showed that vultures and eagles also belonged to the same family called accipitrimorphae (*Jarvis et al., 2014*). The similarity of vulture and eagle is likely attributed to their foraging behaviour as eagles and vultures share a carnivorous diet while chicken is more an omnivore. With uric acid production related to dietary protein concentration and catabolism, it not also surprising that the eagles and vulture would share similar transport mechanism as they would be exposed to the same level of uric acid as an end product of purine metabolism.

In the first step in determining if the evaluated protein was effective, the protein homology results from PHYRE2 were evaluated. For the protein homology, the programme evaluated the similarity of residue probability distributions for each position, secondary structure similarity and the presence or absence of insertions and deletions. For both MRP 2 and 4, the proteins were 57% and 39% similar to bovine MRP 1, with near 100% confidence, supporting the assertion that the channel sequenced where MRP channels. PROTTER and TMHHM results confirmed the two TMDs and NBDs for both proteins. Research on few high-resolution structures of ABC transporters revealed that the former has two TMDs and two NBDs (also known as ABCs) which aligns with our results (*Hollenstein, Dawson & Locher, 2007*; *Oldham, Davidson & Chen, 2008*; *Rees, Johnson & Lewinson, 2009*). The TMDs and NBDs are important for protein functionality, since MRPs are ATP-binding cassettes (ABC) transporters that require the use of ATP to facilitate the movement of an extensive range of substance within the cells (*Linton & Higgins, 1998*). In general for the transport, the TMDs binds the transported substrates and NBDs provide the transport energy by binding and hydrolyzing up to two molecules of ATP (*Newstead et al., 2009*; *Khare et al., 2009*; *Oldham et al., 2007*; *Ward et al., 2007*). Nonetheless some differences have been recorded, with some ABCs transporter that are not specific transporters with energy also being needed to reorient the TMDs, which is also provided by the NBDs (*Jardetzky, 1966*; *Smith et al., 2002*).

Also important in the functionality of the MRP is the presence of the Walker A and B segments which are the ATP binding segments of the NBDs. Comparison of the MRP with their respective human sequence showed a high degree of similarity for both NBDs. With the single amino acid difference, there appears to be no difference in the ATP binding sequence as the difference was at a variable point as reported by *Walker et al. (1982)*. This leads to the conclusion that the ATP binding sites on the protein was functional. Also of importance is the ABC signature motif. The motif was minimally different to that of the human reported sequence as for the Walker segments. This is an important feature as studies have shown that a mutation in this region renders the transporter unable to function. TMHMM and PROTTER predicted the transmembrane helices and the N-glycosylation sites of the latter proteins with MRP2 having 13 and MRP4 having 5 N-glycosylation sites. The importance of glycosylation for the MRP has been evaluated in other studies. It has

been demonstrated that glycosylation is important for the translocation of the MRP from the endoplasmic reticulum to the cell membrane (*Zhang et al., 2005*). In the absence of glycosylation, the transporter remains inactive. Based on this result, we are confident that the evaluated sequence would result in the expression of a viable drug efflux transporter.

The presence of MRP2 and MRP4 in the vulture supports the presence of the described uric acid pathways as for other species which provides the first step for the further evaluation of the mechanism of toxicity. The importance of the channels in toxicity is evident from the study by Konig (*1999*, *2003*) who showed that rat mutants and humans lacking MRP2 gene had compromised excretion of organic anions from the liver leading to mild liver disease and inherited jaundice (*König et al., 1999*; *König et al., 2003*). This finding was supported by an MRP4 functionality study, where *Bakos et al. (1997)* and later *Ren et al. (2004)* demonstrated that the replacement of a highly conserved glycine in NBD1 with an aspartate in vector-transfected cells resulted in intracellular accumulation of the evaluated drugs, confirming their role as efflux transporters. The importance of MRP4 channel was further highlighted in Abcc4 −/− mice study, where the absence of the transporter played a role in acute 9-(2-phosphonylmethoxyethyl) adenine (PMEA) toxicity, suggesting a protective role for MRP4 more in the bone marrow, gastrointestinal tract, thymus and spleen (*Belinsky et al., 2007*). A similar study on humans comparing two antiviral agents, adefovir (PMEA) and azidothymidine (AZT), by measuring the intracellular accumulation, showed a significantly higher intracellular accumulation of PMEA and AZT in cells without the functional allele in comparison to the controls since these cells were unable to extrude the mentioned drugs (*Abla et al., 2008*). In addition to the absence of proper expression playing a role in the intracellular accumulation of excretory substrates, the functioning of the MRP2 and MRP4 may be altered by the NSAIDs such as diclofenac. This is best seen by the excretion of prostaglandins by the kidney being inhibited by probenecid, indomethacin, and PAH which all interact with MRP4 (*Adachi et al., 2002*; *Reid et al., 2003*). Also in a study by *El-Sheikh et al. (2007)*, the excretion of methotrexate by the MRP 2 and 4 was inhibited by numerous NSAIDs including diclofenac, ketoprofen and phenylbutazone all known to be toxic in vultures.

Despite the importance of the result obtained in this study, it should be noted that this is the first step in the process of transporter identification as *silico* analysis is only able to predict the structure and functions of the proteins sequenced (*Punta & Ofran, 2008*; *Barik et al., 2019*). The next step would be to evaluate the functional expression of these transporters. The reason for this is that the sequencing undertaken for this study was only predictive and based on similarity to information published for other species especially man. Also of importance to note is the sample size of two used due to endangered nature of the species evaluated. For the functional studies various technologies may be used. The most commonly available tools would be molecular cloning and transporter studies with specific substrates and inhibitors; immunohistochemistry to identify the location of the transporter in the renal tubules or western blotting to identify protein expression in the kidney (*Haimeur et al., 2004*). These methods do however have disadvantages as with in silico analysis since the assumption is that results are based on conserved physiological function which may not be the case as already evident by the unexpected sensitivity of

old world vultures to diclofenac (*Klein, Sarkadi & Váradi, 1999*; *Punta & Ofran, 2008*; *Naidoo et al., 2007*). Other concerns with western blotting and immunohistochemistry is that the methodology is reliant on antibodies being able to cross react with vulture tissue which might not be the case since most antibodies are designed for rodent or human use. Molecular cloning, while very effective in studying transporter functionality has the inherent risk of the expressed protein being non-functional due to incorrect folding or improper translocation to the cell membrane e.g., when MRP1 was first cloned it did not overexpress P-glycoprotein, while later it was shown to be overexpressed and well conserved in many multidrug resistance protein family (*Mirski, Gerlach & Cole, 1987*; *Cole et al., 1992*; *Krishnamachary & Center, 1993*).

## CONCLUSION

For this study, we show the presence of MRP2 and MRP4 in the kidneys of AWB vultures, supporting the presence of the expected cellular pathways in uric acid excretion. Phylogenetic analysis also confirmed that vulture and eagle are on the same clade in contrast to chicken moreover in slico revealed two nucleotide binding domains (NBDs) and two transmembrane domains (TMDs) for the latter channels with an extra transmembrane domain (TMD) for MRP4. The former confirm that the transporters function the same as other species. Based on this results, further studies can focus on clonal expression of these transporters to ascertain if they are inhibited by the NSAIDs and thereby explain the hyperuricemia associated with diclofenac toxicity in vultures.

### Funding
This work is supported by National Research Foundation, South Africa. The funders had no role in study design, data collection and analysis, decision to publish, or preparation of the manuscript.

### Grant Disclosures
The following grant information was disclosed by the authors:
National Research Foundation, South Africa.

### Competing Interests
Aron Abera is employed by Inqaba Biotechnology.

### Author Contributions
- Bono Nethathe conceived and designed the experiments, performed the experiments, analyzed the data, prepared figures and/or tables, authored or reviewed drafts of the paper, and approved the final draft.
- Aron Abera performed the experiments, analyzed the data, authored or reviewed drafts of the paper, and approved the final draft.

- Vinny Naidoo conceived and designed the experiments, analyzed the data, prepared figures and/or tables, authored or reviewed drafts of the paper, and approved the final draft.

## Animal Ethics

The following information was supplied relating to ethical approvals (i.e., approving body and any reference numbers):

The University of Pretoria Animal Ethics Committee approved this research (V108-16).

## DNA Deposition

The following information was supplied regarding the deposition of DNA sequences:

The golden eaglen MRP2 (XM_030031380.1), MRP4 (XM_030036470.1) and Gyps Himalyanesis (KX168697) and the predicted AWB MRP2 (MN691108) and MRP4 (MN691109) genes are available at GenBank.

## Data Availability

The raw data sequences are available at NCBI (GenBank) SRA: PRJNA560189.

## Supplemental Information

Supplemental information for this article can be found online at http://dx.doi.org/10.7717/peerj.10422#supplemental-information.

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
