# Peer review of "Expression and phylogeny of multidrug resistance protein 2 and 4 in African white backed vulture (Gyps africanus)"

_PeerJ, doi:10.7717/peerj.10422_

## Round 0.1 · original submission · Minor Revisions

Thank you very much for your interesting work. Both reviewers were giving you some advice to complete your manuscript before consider to published in PeerJ. We hope you can address all comments, point by point. I am looking forward to receiving your revision.

Reviewer 1 ·

Basic reporting

This manuscript is interesting and well prepared.

Experimental design

In this study, only 2 vulture samples were collected. One was used for NGS analysis, and another was used for PCR and Sanger sequencing. Could the other add more samples for both NGS and Sanger sequencing to make the data more valid? Alignment of the sequences from both analyses will give the consensus information of the MRPs’ genetic background in vultures.

Validity of the findings

To predict the molecular structure and function of MRPs, the in silico analysis was used in this study. Could the authors give discussion on the molecular functional study of MRPs using other approaches (for example the molecular cloning study, and the gene or protein manipulation)? Advantages and disadvantages of the analyses should be discussed.

Reviewer 2 ·

Basic reporting

No comment

Experimental design

No comment

Validity of the findings

No comment

Additional comments

The manuscript entitled “Expression and phylogeny of multidrug resistance protein 2 and 4 in African white backed vulture (Gyps africanus)” is quite well structured and written. The authors have addressed each of my concerns and I am happy to recommend acceptance of this manuscript in its present form with just some corrections required.

Annotated reviews are not available for download in order to protect the identity of reviewers who chose to remain anonymous.

---

## Round 0.2 · Minor Revisions

Before accepting your manuscript, I need you to address the comments from Reviewer 1.

Reviewer 1 ·

Basic reporting

n/a

Experimental design

n/a

Validity of the findings

Since the prediction of the molecular structure and function of MRPs in this study is based on only 2 vulture samples and relied on the in silico analysis, the reviewer suggests that the authors should give discussion on the molecular functional study of MRPs using other approaches (for example the molecular cloning study, and the gene or protein manipulation). Advantages and disadvantages of the analyses should be discussed comparatively.
The discussion detail will clarify the reliability and the bias issue of the results in this study.

Reviewer 2 ·

Basic reporting

no comment

Experimental design

no comment

Validity of the findings

no comment

Additional comments

The Authors have successfully addressed all reviewer suggestions.

---

## Round 0.3 · accepted · Accept

Congratulation on your publication.